# Effects of Rhapontigenin as a Novel Quorum-Sensing Inhibitor on Exoenzymes and Biofilm Formation of *Pectobacterium carotovorum* subsp. *carotovorum* and Its Application in Vegetables

**DOI:** 10.3390/molecules27248878

**Published:** 2022-12-14

**Authors:** Bincheng Li, Jiaoli Huang, Youjin Yi, Sisi Liu, Rukuan Liu, Zhihong Xiao, Changzhu Li

**Affiliations:** 1College of Food Science and Technology, Hunan Agricultural University, Changsha 410128, China; 2State Key Laboratory of Utilization of Woody Oil Resource, Hunan Academy of Forestry, Changsha 410004, China

**Keywords:** biofilms, exoenzymes, *Pectobacterium carotovorum* subsp. *carotovorum*, quorum-sensing inhibitor, rhapontigenin, keeping vegetables fresh

## Abstract

The aim of this study was to devise a method to protect Chinese cabbage (*Brassica chinensis*) and lettuce (*Lactuca sativa*) from bacterial-disease-induced damage during storage. Thus, the potential of rhapontigenin as a quorum sensing (QS) inhibitor against *Pectobacterium carotovorum* subsp. *carotovorum* (*P. carotovorum*) was evaluated. The QS inhibitory effects of rhapontigenin were confirmed by significant inhibition of the production of violacein in *Chromobacterium violaceum* CV026 (*C. violaceum*, CV026). The inhibitory effects of rhapontigenin on the motility, exopolysaccharide (EPS) production, biofilm formation and virulence–exoenzyme synthesis of *P. carotovorum* were investigated. Acyl-homoserine lactones (AHLs) were quantified using liquid chromatography–mass spectrometry (LC–MS). The inhibitory effects of rhapontigenin on the development of biofilms were observed using fluorescence microscopy and scanning electron microscopy (SEM). A direct-inoculation assay was performed to investigate the QS inhibitory effects of rhapontigenin on *P. carotovorum* in Chinese cabbage and lettuce. Our results demonstrated that rhapontigenin exhibited significant inhibition (*p* < 0.05) of the motility, EPS production, biofilm formation, virulence–exoenzyme synthesis and AHL production of *P. carotovorum*. Additionally, the result of the direct-inoculation assay revealed that rhapontigenin might provide vegetables with significant shelf-life extension and prevent quality loss by controlling the spread of soft-rot symptoms. Consequently, the study provided a significant insight into the potential of rhapontigenin as a QS inhibitor against *P. carotovorum*.

## 1. Introduction

*Pectobacterium carotovorum* subsp. *carotovorum* (formerly called *Erwinia carotovora* subsp. *carotovora*) is considered a well-documented Gram-negative motile plant-pathogenic bacterium that has been reported to cause infectious soft-rot disease in various types of vegetables during post-harvest handling including Chinese cabbage, lettuce, radish, potato, tomato and onion [1]. Moreover, this pathogen is distributed all over the world and possesses extremely strong survivability as it is widespread and able to survive in the soil or plants as saprophytes for more than a year [2]. Additionally, under conditions advantageous to this pathogenic bacteria, *P. carotovorum* has the ability to secrete multiple exoenzymes and produce biofilms, both of which are factors of crucial importance that contribute to its ability to macerate vegetable organs and tissues. Importantly, owing to the production of biofilms, this pathogen is more resistant to physical and chemical treatment, making it more resistant and persistent than free-living bacteria in diverse environments even under harsh conditions. Consequently, chemical antimicrobials need to be used in larger quantities to obtain positive effects, leading to the abuse of chemical antimicrobials and seriously endangering human health [3,4]. In recent years, the post-harvest vegetable spoilage and cross-contamination resulting from *P. carotovorum* has been of vital concern given the substantial economic losses for farmers, producers and customers worldwide [5]. However, there are a lack of studies on effective inhibition strategies to protect vegetables from *P. carotovorum*.

This pathogen can harness the same genes for the virulent infection of diverse hosts and adopt the same mechanisms to induce disease in organisms [6]. One such crucial and universal mechanism of *P. carotovorum* is to mediate the expression of virulence factors and biofilm formation is QS. QS is a regulatory mechanism of information exchange among bacteria. It is common knowledge that bacteria are single-cell organisms. However, using this signal response system, bacteria can synchronize some specific behaviours in their population so that they may complete functions that a single cell cannot complete, similar to multicellular organisms. This mechanism relies on diffusible chemical signalling molecules called autoinducers (AIs); bacteria can utilise these signalling molecules and their receptors to monitor their population density [7]. In the QS system of *P. carotovorum*, the acylated homoserine lactones (AHLs) act as signalling molecules [8] and rely on the ExpI/R signal system. ExpI/R can regulate different mechanisms such as biofilm formation and antimicrobial resistance and still further regulates pathogenicity such as toxins, adhesins, bacteria motility and virulence–exoenzyme synthesis and is essential for the colonization and infection of hosts. More importantly, it has been observed that discrepancies in *P. carotovorum* virulence levels and the transition from acute to chronic infection have been linked with changes in the concentrations of AHLs and in the expression of the QS-regulated genome [9]. Consequently, current research on new methods for targeting QS in *P. carotovorum* is fundamental to developing novel treatment options for inhibiting its virulence factors.

Some studies have proposed that quorum-sensing inhibitors (QSIs), including synthetic and natural compounds, interfere with QS in *P. carotovorum*. In this sense, phytochemicals have been presented as potential novel QSIs [10]. Stilbenoids, as a type of plant phytoalexin, have shown potential to inhibit the microbial QS system and have gained increasing attention; some examples are resveratrol, piceatannol and oxyresveratrol [11,12,13]. Resveratrol as a “model stilbene” has been demonstrated to be an excellent QSI and anti-biofilm agent by playing a promising role in microbial infection control. It has attracted significant attention in recent years [14]. In the present study, we analysed rhapontigenin, which is also a stilbenoid compound belonging to the derivatives of resveratrol and is widely present in various medicinal plants such as rhubarb, rheum tanguticum Maxim. ex Balf. and other medicinal plants in nature [15]. However, no studies on its anti-QS activity have been reported. Herein, the purpose of this study is to characterize the effects of the QSI activity of rhapontigenin on *C. violaceum* CV026 and *P. carotovorum* and the effects of rhapontigenin on *P. carotovorum* motility, EPS production, biofilm formation and exoenzyme activities. The efficacy of rhapontigenin in maintaining the quality of perishable vegetables post-harvest is also explored.

## 2. Results and Discussion

### 2.1. Evaluation of Quorum-Sensing Activity of Rhapontigenin

#### 2.1.1. Screening of Rhapontigenin at Minimum Inhibitory Concentration (MIC)

To determine the MIC of rhapontigenin, a preliminary screening of four concentrations was performed. The selection of the different concentrations was based in the literature [16]. The chemical structure of rhapontigenin is shown in Figure 1A. According to the chemical structure, rhapontigenin belongs to the derivatives of resveratrol (Figure 1B). Resveratrol functions as an excellent QSI and anti-biofilm agent, as has been previously reported [11]. Here, the MIC of rhapontigenin was evaluated using doubling dilution assays with concentrations ranging from 78 to 1250 µg/mL. The results showed that the MICs of rhapontigenin against *P. carotovorum* and *C. violaceum* CV026 were 313 µg/mL and 156 µg/mL, respectively, showing that *C. violaceum* CV026 was more sensitive than *P. carotovorum* to rhapontigenin. Simultaneously, this result corroborates well the findings of Pham, Duong Quang et al. [17] who reported a MIC value of rhapontigenin of 300 μg/mL for *P. carotovorum*. In addition, Jiyang Sheng et al. [12] have reported that the MIC value of resveratrol at 150 μg/mL for *C. violaceum* CV026, and resveratrol, which is based on structural similarities, may have a certain reference value for the MIC value of rhapontigenin of 156 μg/mL for *C. violaceum* CV026. The growth profiles were then verified by applying rhapontigenin (1/2 MIC, 1/4 MIC, 1/8 MIC and 1/16 MIC) to *C. violaceum* CV026 (Figure 1C) and *P. carotovorum* (Figure 1D) for 24 h, respectively. The results showed that treatment with rhapontigenin at sub-MICs produced no inhibitory effects on bacterial growth compared to the control. Thus, according to our results, only rhapontigenin at sub-MICs (1/2 MIC, 1/4 MIC, 1/8 MIC and 1/16 MIC) that did not interfere with cell growth could be used for all the subsequent QS inhibitory experiment analyses. It was important to ensure that anti-QS activity was not affected by antibacterial activity [18]. Therefore, we referred to MIC values of rhapontigenin against *P. carotovorum* at concentrations of 313 µg/mL. In all subsequent experiments, only those concentrations of rhapontigenin that did not interfere with *P. carotovorum* growth were used.

#### 2.1.2. Quorum-Sensing Inhibition of CV026

To determine the QS inhibitory activity of rhapontigenin, selected sub-MICs were evaluated using the diffusion method for the QS inhibition of CV026. Both the circle diameters of the inhibition of violacein on an agar surface and the inhibition extent of violacein production in *C. violaceum* CV026 by rhapontigenin can be observed in Figure 2. Compared to the control, the capacity of rhapontigenin at the four different concentrations to inhibit the production of violacein was concentration-dependent. Of particular note was that the most significant QSI circle was obtained with the maximum tested concentration (78 µg/mL) (Figure 2B, a). In addition, for the other concentrations significant QSI zones were observed except for the minimum tested concentration (10 µg/mL), indicating that rhapontigenin at sub-MICs inhibited violacein production in CV026 (Figure 2B, a–c). On the other hand, as Figure 2A shows, compared to the positive control and considering the same concentrations, furanone C-30 showed the highest inhibition of violacein in CV026 as the QSI zone (35.36 ± 2.82 mm) was the largest. It was particularly notable that the anti-QS circle diameter for rhapontigenin (26.12 ± 1.66 mm) was bigger than that of resveratrol (21.84 ± 1.32 mm). This indicated that the QSI exhibited by rhapontigenin was better than that exhibited by resveratrol. This is important given that QSI activity is crucial for interfering with bacterial signalling and is not antimicrobial [19].

A second screening followed to determine the extent of the capacity of rhapontigenin to inhibit violacein production which was quantified as anti-QS activity against CV026. As Figure 2C shows, a higher concentration was associated with a greater inhibitory effect. Compared to the negative control, the violacein inhibition rates obtained with rhapontigenin treatments at 1/2 MIC, 1/4 MIC, 1/8 MIC and 1/16 MIC were 6.06%, 23.58%, 45.42% and 59.55%, respectively. Of particular note was that rhapontigenin at 1/2 MIC (78 µg/mL), resveratrol (78 µg/mL) and furanone C-30 (78 µg/mL) inhibited violacein production by 59.55%, 52.68% and 88.91%, respectively. According to the results, rhapontigenin treatment considerably decreased the amount of violacein that *C. violaceum* produced when compared to resveratrol–the positive control–at the same tested concentrations. Additionally, following the analysis of the measurements, it was found that rhapontigenin at 1/16 MIC also showed a slight inhibitory effect. In summary, we further showed that rhapontigenin at sub-MICs had significant QSI activity and that the QSI exhibited by rhapontigenin was better than the QSI exhibited by resveratrol.

To our knowledge, this is the first time that rhapontigenin’s anti-QS activity has been described. Other authors have demonstrated that stilbenoid compounds such as resveratrol, piceatannol and oxyresveratrol could effectively control violacein pigment production in *C. violaceum* CV026 [12,13,19]. Thus far, the mechanisms via which stilbenoid compounds inhibit bacterial QS are complicated and incompletely demonstrated. However, some studies have found that this ability may be due to the double bond in the stilbene skeleton which plays a key role in QSI activity [15,20]. In the interim, Pilar Truchado et al. [19] have confirmed that the anti-QS activity exhibited by resveratrol could inhibit the QS systems of *Yersinia enterocolitica* and *Pectobacterium carotovorum* by interfering with AHL synthesis and accelerating the degradation–transformation of AHLs. Other research studies have demonstrated that the QS inhibition mechanism of stilbenes can be expounded at the level of pathogen metabolism. It is influenced both by pathways relieving the oxidative stress of the pathogen and by pathways inhibiting protein synthesis and energy metabolism, thereby inhibiting the expression of genes related to QS [21]. On the other hand, Jiyang Sheng et al. [12] have also reported that the strength of the anti-QS activity of stilbenes may also be attributed to the quantity and position of hydroxyls in the stilbene skeleton, where the hydroxyls at the 3′ and 5′ positions may represent two vital active sites, meaning that if the hydroxyl group is at position 3′ or 5′ in the stilbene skeleton it is more anti-QS active than at position 2′, 4′ or 6′. This may be a possible reason why the anti-QS activity exhibited by rhapontigenin was better than the anti-QS activity exhibited by resveratrol.

### 2.2. Evaluation of Antibiofilm Capacity of Rhapontigenin against P. carotovorum

#### 2.2.1. Effects of Rhapontigenin on Motility, EPS Production and Biofilm Formation of *P. carotovorum*

*P. carotovorum* is a motile plant-pathogenic bacterium and its motility is an important virulence factor that plays a vital role in the initial period of biofilm formation. In view of bacterial motility being related to biofilm formation, bacterial motility such as swimming, swarming and twitching plays a major role in host surface colonization, the spreading of bacteria across the food surface and in biofilm formation [22]. In particular, Hossain et al. [23] have proposed that the capacity of a pathogen to attach to host surfaces is a critical initial stage of biofilm formation because all other bacteria within the biofilm structure absolutely depend on the adhesive interaction between the host surface and the bacterium for their survival. Therefore, for the purpose of studying the initial process of the formation of biofilm, the three types of motilities (swimming, swarming and twitching) inhibited by rhapontigenin at 1/2 MIC were measured (Figure 3).

The motility of bacteria grown with or without rhapontigenin was evaluated (Figure 3A,B). On day 1, the mean swimming, swarming and twitching diameters in the absence of rhapontigenin were 5.78 ± 0.33, 7.48 ± 0.26 and 7.38 ± 0.47 mm, respectively (Figure 3A(a–c)). When treated with rhapontigenin at 1/2 MIC (156 µg/mL), the bacteria formed a colony with diameters of 4.86 ± 0.40, 3.68 ± 0.15 and 2.64 ± 0.29 mm, respectively (Figure 3A(d–f)). Notably, the swarming and twitching diameters in the presence of rhapontigenin were smaller than those of the control. By day 6 (Figure 3B) we found that with the increase in the incubation time the colony size had increased. It could be clearly observed that rhapontigenin at 1/2 MIC caused a significant reduction in swimming, swarming and twitching motility without affecting cell viability; the mean swimming, swarming and twitching diameters in the absence of rhapontigenin were 21.52 ± 1.11 mm, 23.78 ± 2.39 mm and 12.16 ± 1.57 mm, respectively (Figure 3B(a–c)). Following treatment with rhapontigenin, the swimming, swarming and twitching diameters were 10.66 ± 1.51 mm, 12.54 ± 0.97 mm and 8.72 ± 0.40 mm, respectively (Figure 3B(d–f)). It was observed that treatment with rhapontigenin at 1/2 MIC showed significant (*p* < 0.05) inhibition of swimming, swarming and twitching motility compared to the control group. Motility (swimming, swarming and twitching) is associated with biofilm formation, virulence-factor expression and bacterium colonization and plays a fundamental role in pathogenesis, which is also regulated by the QS mechanism. Previous studies showed that the pathogenicity of *P. carotovorum* was decreased by knocking out key genes that regulate flagella such as *mreB*, *flgK* and *hfq* [24]. Moreover, it has also been reported that the inhibition of flagellum synthesis is probably attributed to a loss of motility. Some studies have evidenced that flagella, type I fimbriae and curli are controlled via QS mechanisms [25]. Similarly, Qi Yanjiao et al. found that the *Rheum tanguticum* Maxim. ex Balf. extract could reduce motility phenotypes of *P. carotovorum* through the inhibition of flagellar genes such as *FlgE*, *FliG*, *MotA*, *FlgC* and *FliP* [26]. In this sense, we present evidence indicating that the effect of rhapontigenin at 1/2 MIC, i.e., to inhibit the three types of motilities (swimming, swarming and twitching), could be attributed to an interference with the QS process. Additionally, Figure 3C shows the trends of the motility inhibition rates of bacteria grown with or without rhapontigenin throughout the culture period. As the action time of rhapontigenin at 1/2 MIC increased, the general trend of the effect of the inhibition of motility first increased and then decreased. In particular, as was observed on day 1, the inhibition rates of twitching, swarming and swimming motility were 64.38%, 50.80% and 6.17%, respectively. The twitching motility of *P. carotovorum* showed the best inhibitory effect at 66.30% on day 3. Regarding swarming motility, the best inhibitory effect, 72.32%, was observed on day 2. The best inhibitory effect on swimming motility, 34.54%, was observed on day 5. These results indicated that the inhibition rates of the three types of motilities were highly dependent on rhapontigenin concentrations. However, with the growth of the bacteria the inhibition rates of motility were significantly reduced by day 6. It is possible that the inhibition rates of motility were culture–time dependent.

EPSs are among the most important components of biofilm and have many functions that are also correlated with biofilm formation [23]. Therefore, to evaluate whether rhapontigenin reduced the production of EPSs, the inhibitory effects of rhapontigenin on EPS production were investigated and the results are presented in Figure 4B. The results indicated that treatments with rhapontigenin at 1/16 MIC, 1/8 MIC, 1/4 MIC and 1/2 MIC significantly reduced the production of EPSs by 12.67%, 36.36%, 59.52% and 83.31%. Based on the above results, it is plausible to conclude that the inhibition of EPS synthesis may be attributed to the previously observed reduction in motility. As EPS synthesis is mediated by QS, it could be concluded that the mechanism of action of treatment with rhapontigenin manifested through the inhibition of the QS system, resulting in a reduced synthesis of EPSs or preventing their production.

Biofilms reduce the effectiveness of chemical treatment and lead to physical preservation control failures. The biofilm development of *P. carotovorum* is controlled by the QS mechanism through ExpI/R proteins [22]. The above motility and EPS tests confirmed that rhapontigenin at sub-MICs could effectively inhibit motility and EPS synthesis. Thus, we evaluated whether rhapontigenin inhibited biofilm formation. The results are presented in Figure 4C. Treatments with rhapontigenin at 1/16 MIC, 1/8 MIC, 1/4 MIC and 1/2 MIC reduced biofilm biomass by 1.51%, 32.34%, 66.83% and 91.79%, respectively. As the figure shows, higher concentrations were associated with a greater inhibitory effect. From the aforementioned results it was observed that treatment with rhapontigenin could inhibit motility, interfere with EPS synthesis and decrease biofilm formation without inhibiting growth. It is believed that these results may be attributed to the rhapontigenin treatment’s interference with the QS mechanism.

#### 2.2.2. Effects of Rhapontigenin on Biofilm Formation of *P. carotovorum* Observed Using Fluorescence Microscopy and Scanning Electron Microscopy (SEM)

In view of the promising anti-biofilm capacities of rhapontigenin, biofilm was stained with crystal violet and acridine orange and observed using inverted fluorescence microscopy and SEM (Figure 4). It was found that *P. carotovorum* were capable of forming a strongly adherent and thick biofilm in the control group (Figure 4B(a,f)), as is indicated by the green-stained cells (Figure 4B(a)). The same results were obtained using SEM, which showed that the biofilm formed by a large number of bacteria exhibited a three-dimensional structure (Figure 4B(f)). However, rhapontigenin-treated samples developed weak biofilms (Figure 4B(b–e)). Cracks begin to appear in the biofilm after treatment with 1/16 MIC of rhapontigenin, and as the concentration increased they failed to develop mature biofilms. Even treatment with rhapontigenin at 1/2 MIC showed almost no biofilm formation. A similar result was also obtained via SEM (Figure 4B(g–j)), with SEM images showing the absence of the appearance of biofilm in the presence of rhapontigenin at 1/2 MIC, with only a few dispersed cells remaining attached. The previous test results of rhapontigenin on motility, EPS synthesis and biofilm formation were corroborated by the microscopic analyses, where it could be observed that in rhapontigenin-treated samples the biofilm development of *P. carotovoram* was inhibited.

### 2.3. Effects of Rhapontigenin on AHL Synthesis of P. carotovorum

As signalling molecules for QS, AHLs have been reported to be associated with the regulation of biofilm development and the pathogenesis of *P. carotovorum* [27]. Miller et al. [28] have reported that Gram-negative bacterial pathogens normally synthesise only one or two major AHL signals, also accompanied by a small volume of other AHLs, depending on the classical LuxI/LuxR system. In *P. carotovorum*, the expression of virulence-factor production is predominantly regulated through 3-oxohexanoyl-L-homoserine lactone (3-oxo-C6-HSL)-dependent QS mechanism. Based on the analysed LC–MS data and compared to the retention times of the standards, our group found that the main AHLs released in *P. carotovorum* were 3-oxo-C6HSL and that the retention time was 3.57 min. The result was in line with previous reports. Accordingly, the AHL-synthesis inhibitory capacity of rhapontigenin against *P. carotovorum* was evaluated, and the results are presented in Figure 5. Compared to the control, treatments with rhapontigenin at 1/2 MIC, 1/4 MIC, 1/8 MIC and 1/16 MIC apparently reduced the synthesis of 3-oxo-C6HSL by 92.81%, 45.36%, 10.24% and 2.59%, respectively. These results demonstrated that rhapontigenin prevented *P. carotovorum* from producing AHL. For this reason, rhapontigenin has not yet been documented as a QS-inhibitor. Resveratrol, which resembles rhapontigenin in structure, has been proven to have a QS-inhibitory activity. Similarly, Pilar Truchado et al. [19] found that resveratrol could lessen the pathogenicity of *Yersinia enterocolitica* and *P. carotovorum* by reducing AHL synthesis and accelerating AHL degradation–transformation. On the other hand, resveratrol may inhibit the pathogenesis of *Aeromonas hydrophila* by inhibiting the expression of virulence factors and biofilm formation at sub-inhibitory concentrations by interfering with the expression of QS genes *ahy*I and *ahy*R and interfering with the QS system of *Aeromonas hydrophila* [29]. The quorum-quenching marine bacterium *Bacillus* sp. OA10 was shown by Aparna Anil Singh et al. [30] to be incapable of degrading the acyl homoserine lactone (AHL) of *P. carotovorum*, but instead displayed QSI activity by likely inhibiting AHL synthesis, which in turn inhibited the QS systems of *P. carotovorum*. In this way, rhapontigenin therapy may prevent the QS mechanism from working in *P. carotovorum* by interfering with AHL synthesis or AHL breakdown which would prevent the formation of biofilms.

### 2.4. Effects of Rhapontigenin on Exoenzyme Activities of P. carotovorum

QS leads to the activation of the genes for exoenzyme synthesis in *P. carotovorum* [31]. The activity of rhapontigenin at sub-MICs on exoenzymes cellulase, pectate lyase, polygalacturonase and protease, which are major weapons for *P. carotovorum* to cause disease in hosts, was assessed. As shown in Figure 6, compared to the control, treatment with rhapontigenin caused a significant reduction in exoenzyme activities. Significantly, rhapontigenin at 1/2 MIC almost inhibited each exoenzyme activity 70% more than the control group. The four types of exoenzyme activities decreased with the increase in rhapontigenin concentration. All these results indicated that the exoenzyme activities in the presence of rhapontigenin were inhibited during the incubation stage.

*P. carotovorum*’s two-dimensional difference gel electrophoresis proteome study has reportedly revealed QS-dependent secreted virulence exoenzymes such as cellulase, proteoglycan hydrolase, and pectate lyase [32]. Our findings showed that rhapontigenin at sub-MICs might suppress the activity of exoenzymes without affecting *P. carotovorum* proliferation. As a result, it is thought that the inhibition of enzyme synthesis may be to blame for the decline in enzyme activity. Importantly, the extracellular enzyme synthesis of *P. carotovorum* is controlled by the QS mechanism [33]. In this sense, we present evidence indicating that the effect of rhapontigenin to inhibit the extracellular enzyme could be attributed to an interference in the QS process. Similarly, Zhang et al. [34] found that the extracellular enzymes controlled by QS of *P. carotovorum* and *Pseudomonas fluorescens* were inhibited using hexanal as a preventive measure. The result revealed that hexanal could decrease the pathogenesis of *P. carotovorum* and *Pseudomonas fluorescens* by inhibiting extracellular enzymes activities such as cellulase, xylanase, pectate lyase, polygalacturonase and protease. As a temporary measure, hexanal could provide a protection to vegetables challenged with spoilage strains. Additionally, AS Vasilchenko et al. [35] reported that oak bark (Quercus sp.) extract could decrease the cellulase and protease activity by inhibiting the QS systems of *P. carotovorum* through interfering with AHL synthesis. The effect of oak bark extract treatment at the transcriptomic level is the suppression of the main QS-related genes, ExpI/R. In these studies, the authors reported that the suppression of QS-related genes was responsible for the decline in virulence–exoenzyme activity. In this sense, it might be stated that rhapontigenin acts by inhibiting *P. carotovorum*’s QS system, which reduces the amount of enzyme synthesis.

### 2.5. Effects of QSI Application of Rhapontigenin on P. carotovorum in Vegetables

According to the above test results it was observed that rhapontigenin, as a QSI, could decrease the levels of virulence factors such as motility, EPS synthesis, biofilm formation and exoenzyme activities by inhibiting the QS system of *P. carotovorum*; thus, we investigated whether it may protect vegetables from virulence infection of *P. carotovorum* during storage. As seen in Figure 7, rhapontigenin (at 0 MIC, 1/8 MIC, 1/4 MIC and 1/2 MIC) was applied to Chinese cabbage and lettuce inoculated with *P. carotovorum*. The results showed that, when compared to the control, rhapontigenin could inhibit the virulence of *P. carotovorum* to varying degrees, thereby reducing the decay rate. In particular, rhapontigenin at 1/2 MIC produced the best inhibitory effect, showing two vegetables that almost did not change colour at the inoculation site. Moreover, higher rhapontigenin concentration was associated with a slower tissue decay rate. These results are of interest, as rhapontigenin could significantly inhibit the invasion by *P. carotovorum* of vegetables in practical applications without vegetables developing drug resistance to pathogens. It is believed that rhapontigenin could be used as a novel quorum-sensing inhibitor, which could be applied in the agricultural industry to prevent post-harvest bacterial diseases.

## 3. Materials and Methods

### 3.1. Microbial Isolates and Growth Conditions

*P. carotovorum* was obtained from China General Microbiological Culture Collection Center (Beijing, China) and was isolated from Chinese cabbage. *C. violaceum* CV026 was kindly donated by Prof. Aiqun Jia, Nanjing University of Science and Technology of China, Nanjing, China. Microbial isolates were routinely grown aerobically with shaking in Luria–Bertani medium (LB; Sangon Biotech, Shanghai, China) broth for CV026 and nutrient broth medium (NB; Sangon Biotech, China) for *P. carorovorum.* All strains were incubated at 28 °C and the pH of the culture media was 7.0 unless otherwise specified.

### 3.2. Determination of Minimum Inhibitory Concentration (MIC) of Rhapontigenin

Rhapontigenin (purity ≥ 98%) was purchased from Sigma. Stock solutions were prepared by dissolving Rhapontigenin in dimethyl sulfoxide (DMSO). Strains were routinely grown overnight aerobically at 28 °C on Agar media (NB and LB), and bacterial fluids were prepared by diluting a chosen single colony in liquid media (NB and LB) at 28 °C for 24 h. The bacterium solution (100 µL) was directly added to the liquid medium (100 mL) and adjusted to an inoculum of approximately 1–5 × 10^5^ CFU mL^−1^ (0.05 OD at 600 nm) by hemocytometer method. MIC tests were carried out according to the reference [11] with a few modifications. Briefly, using a tissue culture 48-well microplate, one hundred microlitres of rhapontigenin were added to nine hundred microlitres of diluted bacterial fluid and serial dilutions were performed in the medium to achieve a concentration gradient from 1250 µg/mL to 78 µg/mL. DMSO was used as the negative control. the plates were incubated at 28 °C for 24 h. Antimicrobial activity was detected at OD_600_ using a microplate reader (Biotek Elx800, Winooski, VT, USA).

To obtain growth measurements using a tissue culture 48-well microplate, overnight cultures of *P. carotovorum* (900 µL) and *C. violaceum* CV026 (900 µL) with an inoculum of about 1–5 × 10^5^ CFU mL^−1^ (0.05 OD at 600 nm) were separately added to 100 µL of liquid media (NB and LB) supplemented with different concentrations of rhapontigenin and were then incubated at 28 °C for 24 h. The same volume of DMSO served as the negative control. Growth was determined by measuring OD_600_ using a microplate reader (Biotek Elx800, Winooski, VT, USA) in triplicate.

### 3.3. Quorum-Sensing Inhibition (QSI) Assays

Quorum-sensing inhibition (QSI) assays were performed according to the procedure described in [12] with a few modifications. Briefly, overnight cultures of *C. violaceum* CV026 (100 µL) were added to LB agar plates (100 mL) followed by the AHLs solution (200 µL) being added to LB medium. Next, our group used the well-diffusion method using an Oxford cup to screen the QSI activity of rhapontigenin (30 µL) at different concentrations (1/2 MIC, 1/4 MIC, 1/8 MIC and 1/16 MIC). DMSO served as the negative control. The same volumes of resveratrol (at 78 µg/mL) and furanone C-30 (at 78 µg/mL) served as the positive controls. Finally, the plates were incubated at 28 °C for 24 h.

### 3.4. Quantification of Violacein Production

The quantification of violacein production was performed according to the procedure described in [19] with a few modifications. *C. violaceum* CV026 with an inoculum of about 1–5 × 10^5^ CFU mL^−1^ (0.05 OD at 600 nm) was inoculated in 2 mL of LB in twenty-four-well plates. AHLs were added to the wells that contained different sub-inhibitory concentrations of rhapontigenin, which were then cultured for 24 h; a 1-mL portion of culture from each well was centrifuged at 10,000 rpm for 5 min to precipitate violacein. The obtained pellet was dissolved in 1 mL of DMSO and vortexed vigorously to solubilize the violacein completely. The mixture was centrifuged at 10,000 rpm for 5 min again to remove the bacterial cells by measuring OD_585_ using a microplate reader (Biotek Elx800, Winooski, VT, USA). DMSO served as a negative control. The experiments were performed in triplicate and the percentage of inhibition was calculated using the following formula:(1)$$\begin{array}{c} \underline{\mathrm{control}\;{\mathrm{OD}}_{585}-\mathrm{test}\;{\mathrm{OD}}_{585}}\\ \;\mathrm{control}\;{\mathrm{OD}}_{585} \end{array}\times 100$$

### 3.5. Detection of Motility, EPS Production and Biofilm Formation

Motility (swimming, swarming and twitching) was determined using the methods described in [36]. Rhapontigenin was diluted with motility medium to the concentration of 1/2 MIC. Swimming and swarming plates were inoculated with 2 µL *P. carorovorum* broth culture, representing 1–5 × 10^5^ CFU/Ml (0.05 OD at 600 nm). For swim plates (0.3% agar), the inocula were placed directly into the centre of the agar so that the motility within the semi-solid agar could be evaluated. For swarm plates (0.5% agar), the inocula were placed on the agar surface (centre), enabling visualization of motility across the agar surface. For twitching motility (1% agar), a colony of bacteria was inoculated deep into the agar with a sterile needle so that the needle touched the agar–dish interface and the motility at this interface was subsequently measured. DMSO served as a negative control. The diameters of the swarming, swimming and twitching motility zones were measured after incubation at 28 °C for 1–6 days.

The effects of rhapontigenin at sub-MICs on EPS generation were determined as described in [22]. Using a tissue culture 24-well microplate, overnight cultures of *P. carotovorum* (1900 µL) with an inoculum of about 1–5 × 10^5^ CFU mL^−1^ (0.05 OD at 600 nm) were separately added to 100 µL of NB medium supplemented with different concentrations of rhapontigenin. They were then incubated at 28 °C for 24 h. A 2 mL portion of culture from each well was centrifuged at 6000 rpm for 20 min. Filtered supernatant (0.22 µm) was added to three volumes of chilled ethanol and incubated overnight at 4 °C to precipitate the dislodged EPSs. Precipitated EPSs were collected by centrifugation at 6000 rpm for 20 min at 4 °C. The obtained pellet was dissolved in 1 mL distilled water and stored at −20 °C until further use. DMSO served as a negative control. The total carbohydrate content in the EPSs was quantified by the phenol–sulfuric acid method using glucose as a standard. The inhibition ratio was calculated as follows: Inhibition ratio (%) = 100 × (Total carbohydrate content of control value—Total carbohydrate content of sample value)/Total carbohydrate content of control value.

The microplate assay of biofilm was performed according to the procedure described in [11] with a few modifications. Briefly, using a tissue culture 24-well microplate, overnight cultures of *P. carotovorum* (1900 µL) with an inoculum of about 1–5 × 10^5^ CFU mL^−1^ (0.05 OD at 600 nm) were separately added to 100 µL of NB medium supplemented with different concentrations of rhapontigenin. These were incubated at 28 °C for 24 h. After incubation, planktonic cells were removed and the biofilms were stained with 200 µL crystal violet (0.1%) for 10 min. Excess crystal violet was rinsed off by deionized water and bound crystal violet was solubilized in 300 µL of 95% ethanol. DMSO served as a negative control. Biofilms were quantified by reading the microplates at 570 nm (Biotek Elx800, Winooski, VT, USA). The experiments were performed in triplicate and the percentage of inhibition was calculated using the following formula:(2)$$\begin{array}{c} \underline{\mathrm{control}{\mathrm{OD}}_{570}-\mathrm{test}{\mathrm{OD}}_{570}}\\ \mathrm{control}{\mathrm{OD}}_{570} \end{array}\times 100$$

### 3.6. Observation of Biofilms Using Fluorescence Microscopy and Scanning Electron Microscopy

The observation of biofilms was performed according to an already standardized method [11]. Briefly, NB broth with and without rhapontigenin was inoculated with *P. carorovorum* cultures, representing 1–5 × 10^5^ CFU/mL (0.05 OD at 600 nm), and incubated in twenty-four well plates with chambered cover slides at 28 °C for 24 h.

For fluorescence microscope measurements, the biofilms on the slides were washed three times with PBS to remove the planktonic cells, freeze-dried at −20 °C for 6 h, stained with 0.01% acridine orange for 30 min, and then rinsed with distilled water to remove excess stain. Finally, they were observed using a fluorescence microscope (AxioVert A1) at magnification of 400×.

To obtain scanning electron microscopy (SEM) measurements. biofilms were fixed with 2.5% glutaraldehyde for 10 min and washed with graded ethanol (60% *v*/*v*, 70% *v*/*v*, 80% *v*/*v* and 90% *v*/*v*). Biofilms were subsequently freeze-dried at −20 °C for 6 h, gold-coated, and subjected to SEM (Hitachi TM3030, Tokyo, Japan) at a magnification of 3000×.

### 3.7. LC–MS/MS Analysis of AHLs Produced

The extraction of AHLs was performed as described in [22] with some modifications. Standards of 3-oxohexanoyl-L-homoserine lactone (3-oxo-C6-HSL) (purity ≥ 96%) were purchased from Sigma. Briefly, overnight cultures of *P. carotovorum* (0.05 OD at 600 nm) were inoculated in 50 mL of NB that was supplemented with rhapontigenin at different concentrations (1/2 MIC, 1/4 MIC, 1/8 MIC and 1/16 MIC). After incubation at 28 °C for 24 h at 180 rpm, the supernatant was centrifuged at 10,000 rpm at 4 °C for 15 min, extracted twice with acidified ethyl acetate (50 mL, 0.5% formic acid), concentrated, and dissolved in 5 mL methanol. Finally, the supernatant was passed through a 0.22 µm filter and left on standby at −20 °C until further LC–MS/MS analyses (LCMS-8045; SHIMADZU, Japan). In addition, DMSO was used as a control. The determination of 3-oxo-C6-HSL was carried out via LC–MS. The flow rate was set at 0.3 mL/min. The mobile phase consisted of methanol (buffer A) and water (buffer B). A volume of 10 μL was injected into a column of 2.1 mm × 100 mm × 2 μm. Elution was conducted as follows: 0.0–1.0 min, 20% buffer A; 1.0–6.0 min, 20–55% buffer A; 6.0–9.5 min, 85% buffer A; 9.5–10.0 min, 20% buffer A. Finally, the detection of AHLs was based on the peak time of the standard and ion fragmentation of the standard reference substance, and the characteristic ion fragmentation was *m/z* 102. The final results of the test were integrated with the peak area of the AHLs.

### 3.8. Detection of Rhapontigenin on Exoenzymes

The extracellular activity (cellulase, protease, pectate lyase and polygalacturonase) of *P. carotovorum* was examined according to an already standardized method [34].

Briefly, to obtain protease measurements, a sterile supernatant (500 µL) of *P. carotovorum* was mixed with 1.0% azocasein (5.5 mL, Sangon Biotech, China) in 1 M Tris-HCl (pH = 8.5). The mixture was incubated at 40 °C for 1 h. The undigested substrate was precipitated by adding 4.0 mL of 10% trichloroacetic acid for 20 min, followed by a 10-min centrifugation at 10,000 rpm. The supernatant was collected and supplemented with an equal volume of 1 M NaOH. Protease activity was measured at 436 nm (A436) with an ultraviolet spectrophotometer. In addition, DMSO was used as a control. The activity was calculated as ΔA436/min/mL, and the background activity of a medium control was subtracted.

Briefly, to obtain pectate lyase measurements, a sterile supernatant (500 µL) of *P. carotovorum* was mixed with 9500 µL of Pel reaction mix [100 mM Tris–HCl (pH 8.5), 0.5 mM CaCl2 and 0.5% polygalacturonic acid]. The mixture was incubated at 50 °C for 30-min followed by a 10-min centrifugation at 10,000 rpm. The undigested substrate was mixed by adding 2500 µL water and 5000 µL 3,5-dinitrosalicylic acid. It was then reacted in a boiling water bath for 5 min followed by a 10-min centrifugation at 10,000 rpm. The supernatant was collected and measured at 540 nm(A540) with an ultraviolet spectrophotometer. In addition, DMSO was used as a control. The activity was calculated as ΔA540/min/mL, and the background activity of the medium control was subtracted.

Briefly, to obtain polygalacturonase measurements, a sterile supernatant (500 µL) of *P. carotovorum* was mixed with 2500 µL of reaction mix [50 mM sodium acetate (pH 5.5) and 0.5% polygalacturonic acid]. The mixture was incubated at 28 °C for 30 min. The undigested substrate was mixed by adding 2500 µL water and 5000 µL 3,5-dinitrosalicylic acid. It was then reacted in boiling water bath for 5 min followed by a 10-min centrifugation at 10,000 rpm. The supernatant was collected and measured at 540 nm (A540) with ultraviolet spectrophotometer. In addition, DMSO was used as a control. The activity was calculated as ΔA540/min/mL, and the background activity of a medium control was subtracted.

Briefly, to obtain cellulase measurements, a sterile supernatant (500 µL) of *P. carotovorum* was mixed with 2000 μL of reaction mix [2.5 mg/mL carboxymethyl cellulose, 0.2 M phosphate buffer]. The mixture was incubated in a water bath at 42 °C for 1 h. The undigested substrate was precipitated by adding 7.5 mL ethanol-acetone (volume ratio 2:1) for 20 min followed by a 10-min centrifugation at 10,000 rpm. The supernatant was collected and measured at 550 nm (A550) with an ultraviolet spectrophotometer. In addition, DMSO was used as a control. The activity was calculated as ΔA550/min/mL, and the background activity of the medium control was subtracted.

### 3.9. QSI Application of Rhapontigenin in Vegetables

Chinese cabbage and lettuce were used for the in vivo assays aimed at establishing whether QSI by rhapontigenin could inhibit infection in Chinese cabbage and lettuce. Our group used the method reported in [37] with some modifications. Briefly, overnight cultures of *P. carotovorum* were added to 50 mL of NB medium, incubated at 28 °C at 180 rpm for 24 h and diluted to 1 × 10^6^ CFU/mL (McFarland scale 0.5). Chinese cabbages and lettuce were purchased from a local market in Changsha, China, and washed thoroughly with running tap water. Damaged outer leaves were removed and discarded. Fresh, intact cabbages and lettuce were cut with a sterile scissor into 4 × 4 cm^2^. Prior to inoculation, each sliced sample was soaked in 10,000 ppm sodium hypochlorite (NaOCl; Sangon Biotech, China) and 75% ethanol (99% fermented ethanol, anhydrous; Sangon Biotech, China) for 30 s each. A hole was made with a sterile needle on each slice that was subsequently inoculated with 5 µL of *P. carotovorum* suspension. The inoculated samples were left at 25 °C for 1 h until they were completely dry so that the bacteria could efficiently attach. Subsequently, 10 µL of rhapontigenin containing various concentrations (1/2 MIC, 1/4 MIC and 1/8 MIC) was applied dropwise to the injection site. In addition, DMSO was used as a control. The inoculated samples were kept in sterile Petri dishes. The lesion areas of Chinese cabbage (4 × 4 cm^2^) and lettuce (4 × 4 cm^2^) were examined after incubation at 28 °C for 1–5 days. This experiment was independently conducted three times with three replicates in each trial.

### 3.10. Statistical Analysis

All the experiments were repeated at least three times and the results were reported as means ± standard deviations (SDs). Graphs were constructed using Origin 95 software (OriginLab, Northampton, MA, USA). The one-way analysis of variance (ANOVA) was employed using SPSS 18.0 software (SPSS, Inc., Chicago, IL, USA) for comparing differences among groups, followed by the Tukey–Kramer test. A *p*-value ≤ 0.05 was considered statistically significant.

## 4. Conclusions

In this study, rhapontigenin was first confirmed to have anti-QS potential in bacteria. The results showed that rhapontigenin could significantly inhibit violacein production (*p* < 0.05) in *C. violaceum* CV026 at sub-MICs. In addition, rhapontigenin was shown to significantly affect the virulence factors of *P. carotovorum* including motility, EPS synthesis and biofilm biomass. Observations using fluorescence microscopy and SEM confirmed that rhapontigenin also effectively inhibited biofilm formation. This provides insights into the possible role of rhapontigenin in the interruption of QS. Interestingly, the QSI application of rhapontigenin for vegetable preservation may inhibit the infection of the virulence exoenzymes of *P. carotovorum* including cellulase, pectate lyase, polygalacturonase and protease, which represent major weapons for the QS-regulated infection by *P. carotovorum*. Therefore, reduced levels of these factors are necessary to inhibit QS. Our present findings showed that virulence–exoenzyme activities decreased with the increase in rhapontigenin concentration and that there were significant differences among the effects of different concentrations of rhapontigenin. Furthermore, our present findings showed that, compared to the control, rhapontigenin allowed the shelf-life of Chinese cabbage and lettuce to be significantly extended and prevented quality loss by controlling the spread of soft-rot symptoms. Furthermore, higher rhapontigenin concentrations were associated with lower the tissue decay rates. There is further potential for rhapontigenin treatment to be used as an alternative to chemical bactericides in the preservation step of post-harvest treatment.

## Figures and Tables

**Figure 1 molecules-27-08878-f001:**
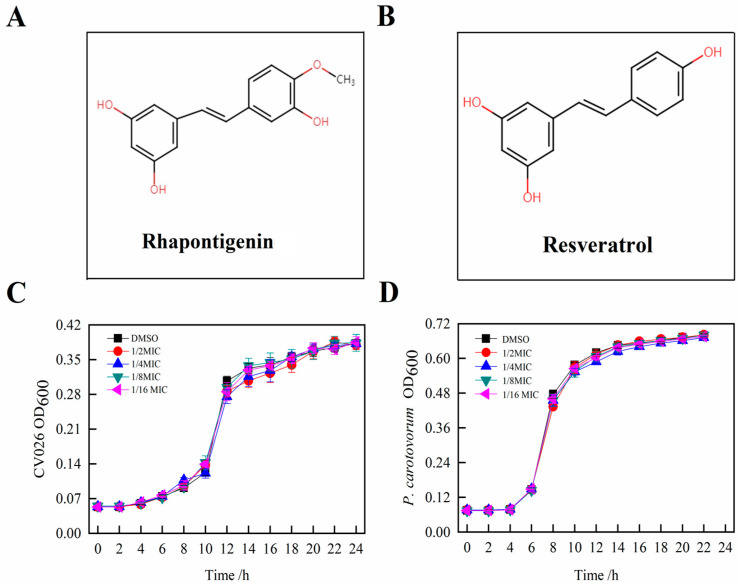
Chemical structure of rhapontigenin (**A**); Chemical structure of resveratrol (**B**); Effects of rhapontigenin on *C. violaceum* CV026 growth (**C**) using 1/2 MIC—78 µg/mL; 1/4 MIC—39 µg/mL; 1/8 MIC—20 µg/mL; and 1/16 MIC—10 µg/mL; and effects of rhapontigenin on *P. carotovorum* growth (**D**) using 1/2 MIC—156 µg/mL; 1/4 MIC—78 µg/mL; 1/8 MIC—39 µg/mL; and 1/16 MIC—20 µg/mL. DMSO served as the negative control. Error bars demonstrate the standard deviations of three measurements.

**Figure 2 molecules-27-08878-f002:**
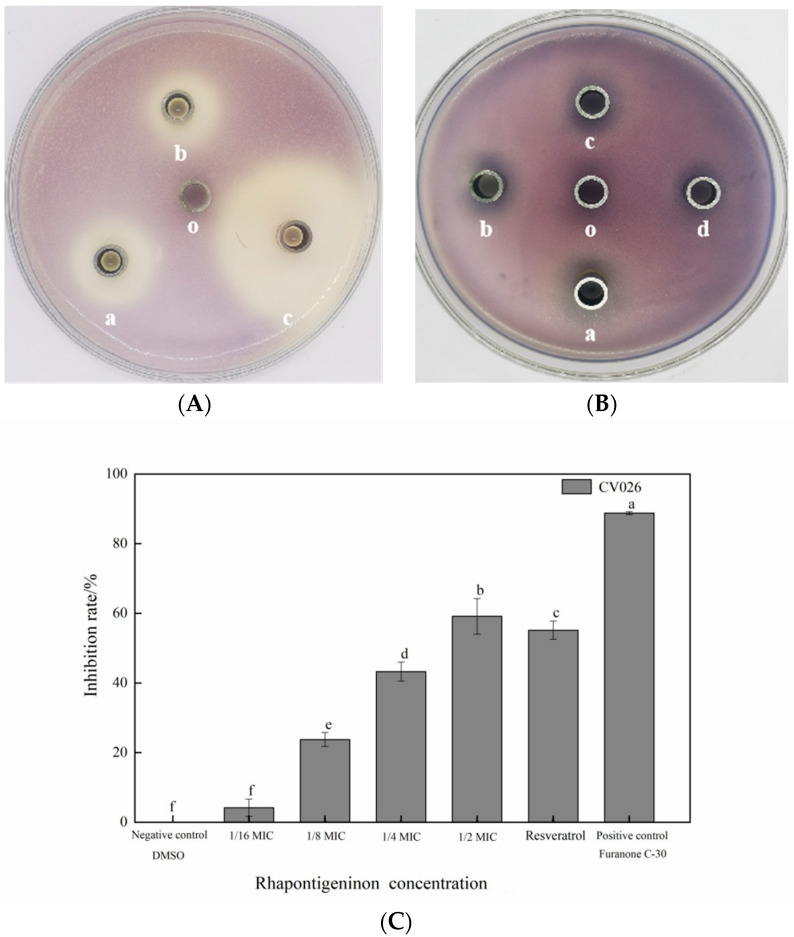
Quorum-sensing inhibitory effects of rhapontigenin on *C. violaceum* CV026 (**A**,**B**); (**A**) a—rhapontigenin (78 µg/mL); b—resveratrol (78 µg/mL); c—furanone C-30 (78 µg/mL); o—DMSO; (**B**) a—rhapontigenin (78 µg/mL); b—rhapontigenin (39 µg/mL); c—rhapontigenin (20 µg/mL); d—rhapontigenin (10 µg/mL); o—DMSO. Inhibition of violacein production in *C. violaceum* CV026 by rhapontigenin; and (**C**): 1/2 MIC—78 µg/mL; 1/4 MIC—39 µg/mL; 1/8 MIC—20 µg/mL; 1/16 MIC—10 µg/mL. Resveratrol (78 µg/mL), furanone C-30 (78 µg/mL) and DMSO served as the negative control. Error bars are labelled with different letters indicate significant differences at *p* < 0.05.

**Figure 3 molecules-27-08878-f003:**
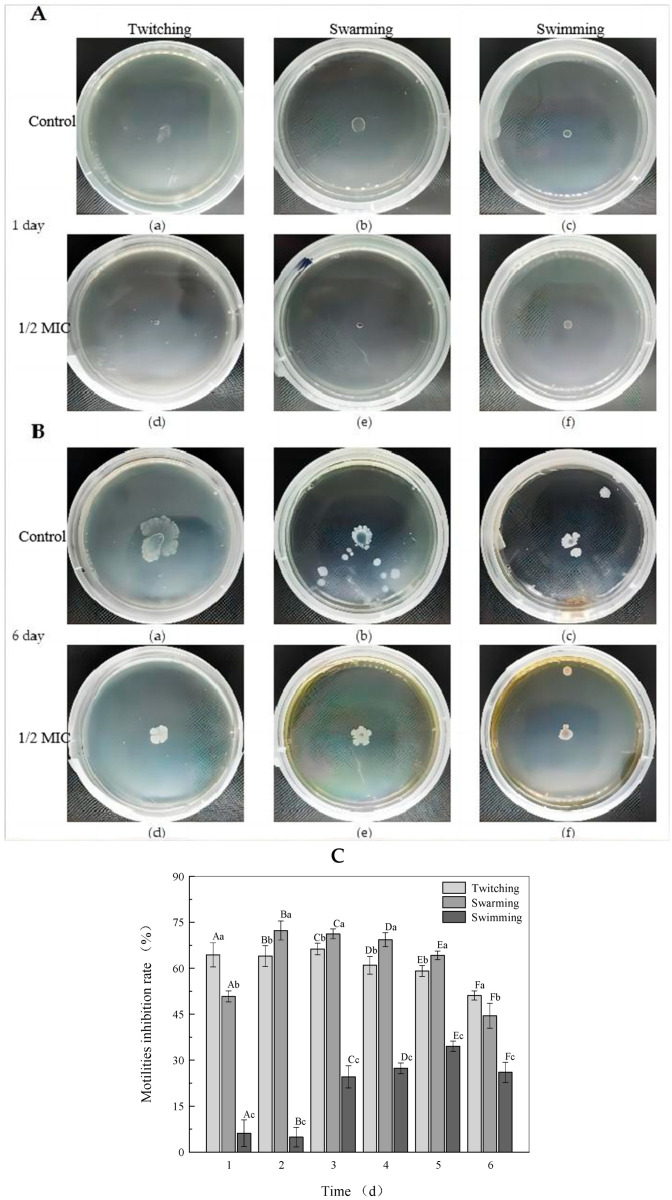
Effects of rhapontigenin at 1/2 MIC on the swimming, swarming and twitching motility of *P. carotovorum* on day 1 (**A**); twitching (a,d), swarming (b,e) and swimming (c,f); DMSO (a–c) and rhapontigenin (d–f). Effects of rhapontigenin at 1/2 MIC on the swimming, swarming and twitching motility of *P. carotovorum* on day 6 (**B**); twitching (a,d), swarming (b,e) and swimming (c,f); DMSO (a–c) and rhapontigenin (d–f); and effects of rhapontigenin on inhibition rate of motility trends of *P. carotovorum* on days 1–6 in the culture period (**C**). Error bars are labelled with different letters indicate significant differences at *p* < 0.05.

**Figure 4 molecules-27-08878-f004:**
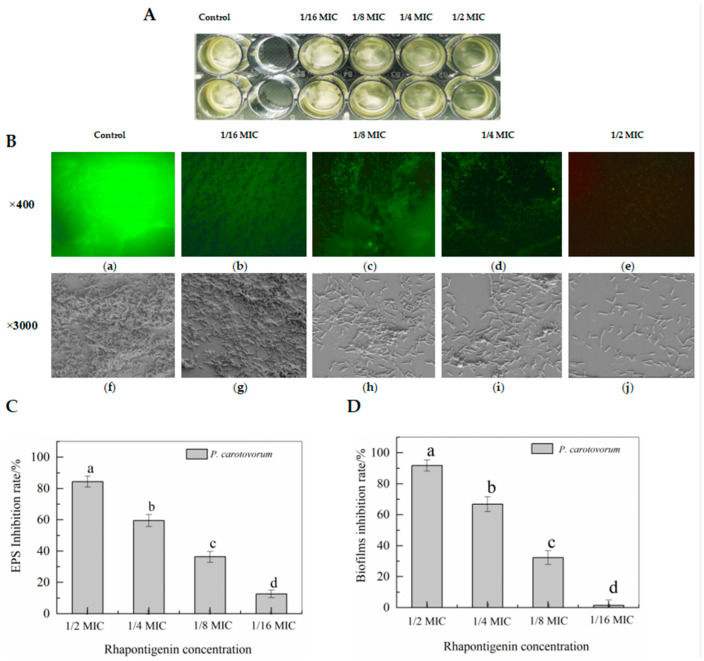
Biofilm biomass of bacterial broth culture co-cultured with rhapontigenin (**A**); Observation of rhapontigenin on biofilm formation using fluorescence microscopy (**B**, a–e) and SEM (**B**, f–j): DMSO (a,f); 1/16 MIC—20 µg/mL (b,g); 1/8 MIC—39 µg/mL (c,h); 1/4 MIC—78 µg/mL (d,i); 1/2 MIC—156 µg/mL (e,j); Effects of rhapontigenin on inhibition rate of EPS synthesis (**C**); and Effects of rhapontigenin on inhibition rate of biofilm formation (**D**). Error bars are labelled with different letters indicate significant differences at *p* < 0.05.

**Figure 5 molecules-27-08878-f005:**
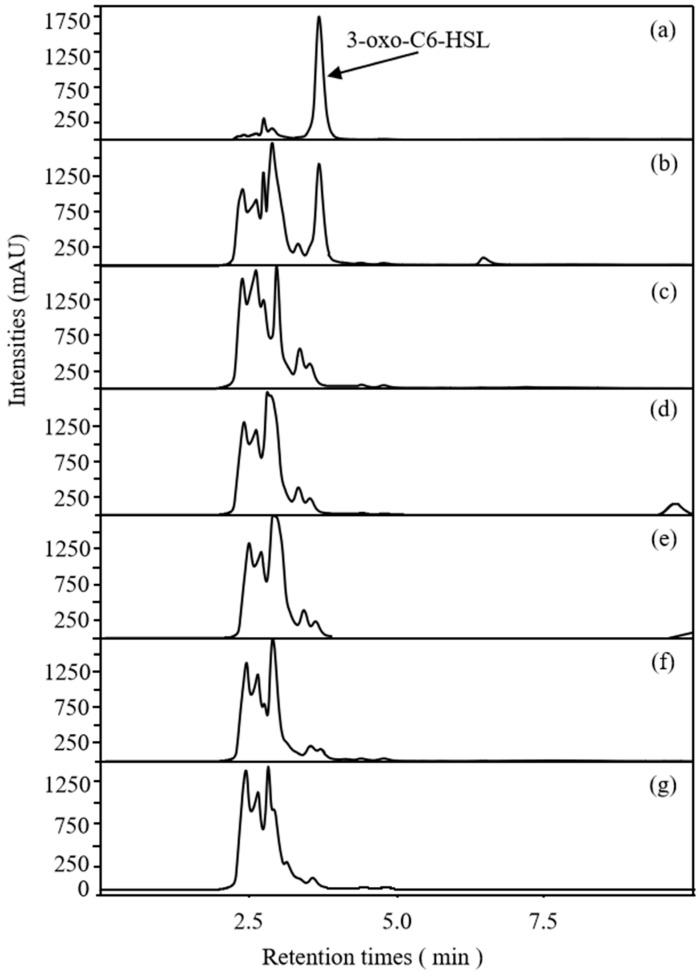
*LC*/*MS* chromatograms of AHLs inhibitory effects of rhapontigenin on *P. carotovorum*. 3-oxo-C6-HSL standard (a); NB + 3-oxo-C6-HSL standard (b). NB + DMSO + *P. carotovorum* strain (c); NB + DMSO + 1/16 MIC (20 µg/mL) of rhapontigenin + *P. carotovorum* strain (d); NB + DMSO + 1/8 MIC (39 µg/mL) of rhapontigenin + *P. carotovorum* strain (e); NB + DMSO + 1/4 MIC (78 µg/mL) of rhapontigenin + *P. carotovorum* strain (f); and NB + DMSO + 1/2 MIC (156 µg/mL) of rhapontigenin + *P. carotovorum* strain (g).

**Figure 6 molecules-27-08878-f006:**
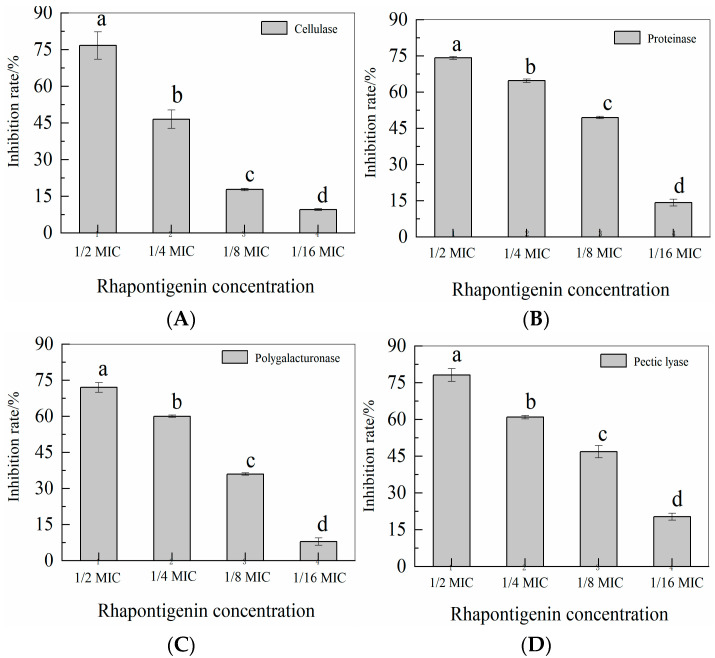
Effects of rhapontigenin on cellulase (**A**); Effects of rhapontigenin on protease (**B**); Effects of rhapontigenin on polygalacturonase (**C**); Effects of rhapontigenin on pectate lyase (**D**); Rhapontigenin dosage: 1/16 MIC—20 µg/mL; 1/8 MIC—39 µg/mL; 1/4 MIC—78 µg/mL; 1/2 MIC—156 µg/mL. Error bars are labelled with different letters indicate significant differences at *p* < 0.05.

**Figure 7 molecules-27-08878-f007:**
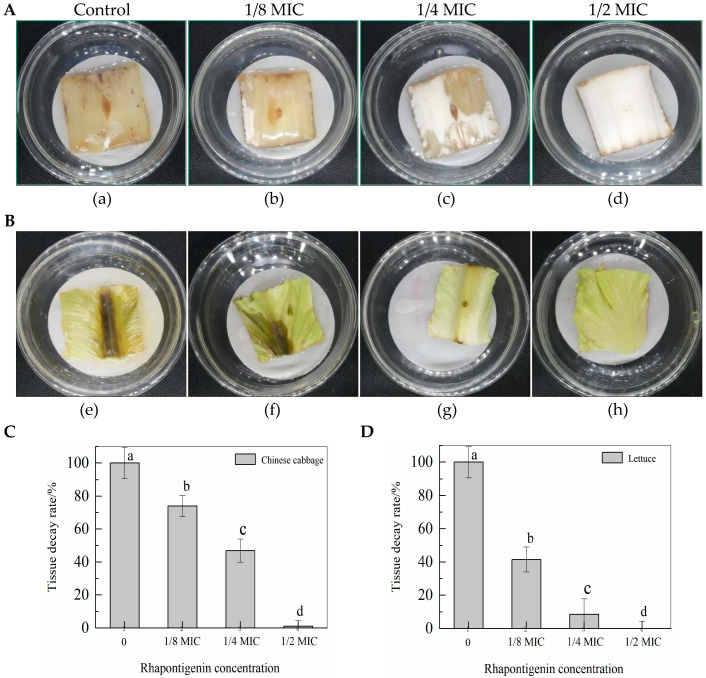
QSI effects of rhapontigenin on *P. carotovorum* virulence infection in Chinese cabbage (**A**): a—DMSO; b—1/8 MIC (39 µg/mL); c—1/4 MIC (78 µg/mL); d—1/2 MIC (156 µg/mL); QSI effects of rhapontigenin on *P. carotovorum* virulence infection in lettuce (**B**); e—DMSO; f—1/8 MIC (39 µg/mL); g—1/4 MIC (78 µg/mL); h—1/2 MIC (156 µg/mL). Tissue decay rate of Chinese cabbage following rhapontigenin administration (**C**); and tissue decay rate of lettuce following rhapontigenin administration (**D**). Error bars labelled with different letters indicate significant differences at *p* < 0.05.

## Data Availability

Not applicable.

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
