# Peer review of "Effects of Rhapontigenin as a Novel Quorum-Sensing Inhibitor on Exoenzymes and Biofilm Formation of Pectobacterium carotovorum subsp. carotovorum and Its Application in Vegetables"

_molecules, 2022, doi:10.3390/molecules27248878_

Round 1

Reviewer 1 Report (Previous Reviewer 3)

The manuscript is much better after major revision. However, some concerns should be addressed before the editor makes a decision on this work.

1. It is no need present the 3-oxo-C6-HSL standard arrows in each row of Figure 5.

2. In Figure 5, are the singles UV absorption from HPLC or ion source density from MS?

Author Response

Reviewer 2 Report (Previous Reviewer 2)

Seems acceptable in current form. 

Author Response

Reviewer 3 Report (Previous Reviewer 1)

The Authors correctly amended the manuscript.

Consequently, I accept the work.

In the subsection “2.1. Strains and Growth Conditions” I advise the Authors to use “microbial isolates” instead of “strains” where it is written.

Author Response

This manuscript is a resubmission of an earlier submission. The following is a list of the peer review reports and author responses from that submission.

Round 1

Reviewer 1 Report

The Authors should use a correct English language. In some cases, the phrases are not understandable. The quality of the English style and grammar is not suitable for a scientific Journal and the manuscript cannot be accepted.

Then, I advise the Authors that the manuscript be revised by a native English speaker to make the text accurate and formal.

Reviewer 2 Report

1 English should be improved. Grammarly errors could be found throughout the paper. 

2 The biosafety of using rhapontigenin on Chinses brassica pekinensis and greengrocery is a must concern.

Reviewer 3 Report

In this manuscript of "Effect of Rhapontigenin as a Novel Quorum Sensing Inhibitor on Exoenzymes and Biofilm Formation of Pectobacterium carotovorum subsp. carotovorum and its Application in Vegetables", the authors identified rhapontigeninon had anti-QS potential to bacterial. Rhapontigeninon was shown that significantly affected virulence factors of P. carotovorum, including motility, EPS synthesis and biofilm biomass. Furthermore, fluorescence microscopy and SEM confirmed that rhapontigeninon effectively inhibited the biofilm formation. The inhibit infection of virulence exoenzymes for P. carotovorum was applied to vegetables preservation.

The manuscript is interesting, and well written. However, the following concerns should be addressed before the editor makes a decision on this work.

1. Through the manuscript, all the evaluations were performed toward P. carotovorum, expect section 3.1.2 Quorum Sensing Inhibition of CV026. You should add the results about the QSI for P. carotovorum, and make some discussion.

2. Please double check Figure 5c, it should be blank without any 3-oxo-C6-HSL, if DMSO the blank control.

3. In Figure 2B note, add the compound name rhapontigeninon.

4. In Figure 6, it is non-sense for the unit (mg/mL) when you apply the MICs at here.

5. Please double check the typos through the manuscript, such as the bacterial names should be italicized; in line 139, the unit of the centrifuged rate; in line 176,  "which resveratrol function as....", in line 315, the production of EPS; in line 377, the serial order is g instead of j; in line 396, "it is believed.....".

Reviewer 4 Report

Dear Authors, in my opinion, this manuscript should not be reviewed before its rewritting and a deep linguistic revision.

Many sentences are difficult to understand, which affects the understanding of the work and the overall assessment of the study.

For example, even the name of the principal compound is rhapontigenin or rhapontigeninon. Which is right?

Moreover, DMSO serves as a negative control of the bacterial culture in one experiment, but allows the bacteria to grow in another. Maybe I did not understand it properly or it is a general lack of understanding of the article due to the low quality of the language.

Round 2

Reviewer 1 Report

In the second review the Authors correct the English style and the grammar, and the text appears clearer and more considerable.

Nevertheless, I have some questions and some points that Authors should clarify.

Minor Points:

Line 64: “Relies” or rely? Are the Authors referring to the AHLs?

Line 67: Correct “adhesions” to “adhesins”

Line 80: “model stilbene”. What do the Authors mean? Maybe stilbenoid?

Line 96: The use of “microbial isolates” is preferred.

Figure 2: Why does the same concentration of rhapontigenin (Aa and Ba) in the 2 different plates prepared with the same method give different results?

Major Points:

-          2.3, 2.4., 2.5 2.6, 2.8, 2.9: The Authors should describe the method without delegating to the reading of other manuscripts. The references of the methods should be added but only after being explained. Moreover the method 2.2 is not well exposed, can the Authors make it clearer?

-          What do the Authors mean with “triplicate”? In all the methodological procedures, how many times do the Authors perform the experiments? I recommend doing at least three different experiments (not the technical replicates) show biological variations indicating the reproducibility of the effect that the Authors are studying.

Reviewer 3 Report

The authors have done all the necessary correction and the manuscript could be accepted in its current version.

Reviewer 4 Report

The authors improved the linguistic quality of the article, mostly. However, there are still a number of issues that need to be corrected before the manuscript is accepted for publication. Moreover, there are still some parts that need profound correction due to writing carelessness and linguistic matter.

Major:

The units that MIC values are normally given is either mg/l or µg/ml. Why did you choose different units?

Line 108 – “at 0.1% (0.5 OD at 600 nm)” – it is not clear for the readers how the suspension was prepared. Did you calculate % of bacteria inoculum density? If so, it is wrong because bacteria are not chemical compounds.

Line 110 and 117 – what does the “amount” mean?

Line 110, 118, 125 – “negative control” of what?

Line 149 – “control” of what?

Line 149 – “3-oxo-C6-HSL” is not mentioned before, thus, its presence here is very confusing.

Line 166 – how did you confirm/calculate/check this inoculum density?

Line 242-244 – this conclusion is an exaggeration; you checked only the selected concentrations of the mentioned compounds.

Line 253 - Erwinia carotovora is the same as Pectobacterium carotovorum which makes the sentence confusing for the readers, suggesting that a different bacteria species have been investigated in this paper.

Line 314-317 – this observation should be discussed/explained in this section

Figure 4 – what is put between control and 1/16 MIC on the plate?

Lines 384-399 and 423-445 need profound correction due to linguistic matter and writing carelessness.

Figures/pictures/graphs need to be changed, most of them is hardy readable.

The first sentence of the Conclusions is basically not your conclusion.

Minor:

Keywords are not listed in an alphabetic order.

Gram is a surname, capital letter is needed (line 36).

Line 63 – repeat/italics.

Line 160, 162, 239, 248, 433, 453 – italics.

Line 203, 229 – “negative control/s” of what?

Line 261 – “two vital active sites” is not clear for me.

Figure 5 – units are missing.

Line 425, 484 – “pectate lyase”

QS abbreviation is introduced 4 times in the manuscript.
